# Iron in Vascular Calcification: Pro-Calcific Agent or Protective Modulator?

**DOI:** 10.3390/ijms262010210

**Published:** 2025-10-20

**Authors:** Enikő Balogh, Andrea Tóth, Viktória Jeney

**Affiliations:** Vascular Pathophysiology Research Group, Research Centre for Molecular Medicine, Faculty of Medicine, University of Debrecen, 4032 Debrecen, Hungary; balogh.eniko@med.unideb.hu (E.B.); andrea.toth@med.unideb.hu (A.T.)

**Keywords:** iron, vascular calcification, valve calcification, vascular smooth muscle cell, valve interstitial cell, osteochondrogenic differentiation, chronic kidney disease

## Abstract

Vascular calcification is a complex, regulated process characterized by the pathological deposition of calcium phosphate minerals in the vascular wall, contributing to cardiovascular morbidity and mortality, particularly in patients with chronic kidney disease (CKD), diabetes, and aging. Once thought to be a passive degenerative process, it is now recognized as an active, cell-mediated phenomenon that shares molecular features with bone formation. Beyond traditional risk factors such as hyperphosphatemia and inflammation, disturbances in iron metabolism have recently emerged as modulators of vascular calcification. Iron, a vital trace element involved in numerous cellular functions, exhibits a dual role as both a potential driver and inhibitor of calcification, depending on its dose, distribution, and cellular context. In this review, we summarize in vitro and in vivo studies investigating the impact of iron on the osteochondrogenic differentiation and calcification of vascular smooth muscle cells and valve interstitial cells. We further highlight mechanistic insights that may explain the divergent findings reported in the literature. Finally, we compile clinical evidence linking disturbances in iron metabolism with coronary artery calcification and cardiovascular mortality in CKD patients.

## 1. Introduction

Vascular calcification is a widespread and clinically consequential process in which calcium phosphate mineral precipitates accumulate within the cardiovascular system, contributing to arterial stiffening, valvular dysfunction, and an increased burden of cardiovascular events. Once regarded as a passive, degenerative sequela of aging, calcification is now understood as an actively regulated, cell-mediated phenomenon that shares molecular hallmarks with bone formation and involves osteochondrogenic reprogramming of vascular smooth muscle cells (VSMCs) and valvular interstitial cells (VICs).

Recent studies addressed the role of iron in modulating cardiovascular calcification (reviewed in [1,2,3]), demonstrating that systemic or local iron excess can promote oxidative stress, lipid peroxidation, and ferroptotic cell death, thereby exacerbating medial calcification and upregulating osteochondrogenic programs in vascular tissue. Conversely, other investigations report that iron excess may attenuate calcification, either by enhancing antioxidant defense systems or by forming soluble iron–phosphate complexes that lower the availability of free inorganic phosphate for pathological calcification.

Vascular calcification is a frequent and severe complication of chronic kidney disease (CKD), strongly contributing to cardiovascular morbidity and mortality. Anemia is another common manifestation of CKD, arising mainly from reduced erythropoietin production and disordered iron metabolism. Iron supplementation, either oral or intravenous, remains a cornerstone of anemia management in CKD patients [4]. Clinical studies have linked iron supplementation to increased survival in CKD populations, suggesting that iron may affect vascular health independently of its role in erythropoiesis. More focused clinical studies have now reported associations between iron markers and measures of coronary calcification: lower serum iron (Fe ≤ 72 µg/dL) and transferrin saturation (TSAT ≤ 21%) have been associated with higher coronary artery calcium scores (CACS ≥ 400) and with worse long-term survival in hemodialysis cohorts; at the same time, absolute iron deficiency (TSAT < 20%, serum Fe < 60) has also been linked with increased cardiovascular mortality in maintenance dialysis patients [5].

Given the clinical importance of calcification in CKD, the need for iron supplementation in these patients, and the increasing recognition of iron biology as a potential determinant of vascular outcomes, a careful synthesis of the clinical associations and mechanistic data is warranted. In this paper, we review and integrate the available clinical and experimental evidence on the effect of iron on cardiovascular calcification, with emphasis on findings in hemodialysis populations. We also discuss the possible cellular mechanisms responsible for the pro-and anti-calcification effects of iron.

## 2. Cardiovascular Calcification

Cardiovascular calcification is a complex and heterogeneous pathological process characterized by the deposition of calcium phosphate crystals within the cardiovascular system. Traditionally viewed as a passive, degenerative consequence of aging, cardiovascular calcification is now recognized as an actively regulated, cell-mediated phenomenon with distinct subtypes, mechanisms, and clinical implications [6,7,8].

### 2.1. Types of Cardiovascular Calcification

The three most commonly described forms of cardiovascular calcifications are intimal (atherosclerotic), medial (Mönckeberg’s), and valve calcification, which are implicated in cardiovascular morbidity and mortality.

Intimal calcification occurs within atherosclerotic plaques and is closely associated with traditional cardiovascular risk factors, including dyslipidemia, hypertension, smoking, and diabetes [9,10]. The calcified deposits in the intima can vary in size and distribution, ranging from microcalcifications to large sheets of calcified tissue [11,12]. Microcalcifications, in particular, have been implicated in plaque rupture by serving as focal points for increased mechanical stress [11]. In contrast, larger, more confluent calcified plaques are thought to contribute to plaque stability but are also associated with increased arterial stiffness and impaired vascular compliance [13]. Overall, intimal calcification is a hallmark of late-stage atherosclerosis, and the coronary artery calcium score is a strong predictor of coronary heart disease events, even stronger than the Framingham risk index [10,14,15].

Medial arterial calcification is characterized by the deposition of calcium in the elastic lamina of the arterial media, leading to increased vascular stiffness and isolated systolic hypertension [16,17]. Unlike intimal calcification, medial calcification occurs independently of lipid accumulation and is more closely associated with aging, diabetes mellitus, and CKD [18,19,20]. This type of calcification is often underdiagnosed due to the absence of luminal narrowing, yet it significantly impairs arterial compliance and increases pulse wave velocity [13].

The most prevalent form of valve calcification is calcific aortic valve disease (CAVD), which is a leading cause of aortic stenosis in the elderly. Valve calcification is characterized by the progressive calcification of aortic valve leaflets, leading to reduced leaflet mobility and increased transvalvular gradients [21]. The pathogenesis of CAVD shares similarities with atherosclerosis, including endothelial dysfunction, lipid infiltration, and inflammatory cell recruitment, but also involves unique VIC phenotypes and mechanical stress-related pathways [22].

### 2.2. Calcification Is an Actively Regulated Process

Vascular calcification is a dynamic, cell-regulated process that resembles bone formation, involving multiple molecular and cellular pathways. In particular, VSMCs and VICs undergo cell lineage reprogramming toward osteochondrogenesis in response to specific pro-calcific stimuli, resulting in the deposition of hydroxyapatite crystals within the vascular wall or the valve, respectively. This transformation reflects an imbalance between calcification inducers and inhibitors, and is modulated by inflammatory signals, metabolic changes, and mechanical stressors.

#### 2.2.1. Osteochondrogenic Transcription Factors Regulate Cardiovascular Calcification

The osteochondrogenic transdifferentiation of VSMCs and VICs is governed by a complex network of transcription factors that normally dictate osteogenesis, chondrogenesis, and matrix mineralization. These include Runt-related transcription factor 2 (Runx2), a widely recognized master regulator of bone differentiation; Osterix, a critical transcription factor downstream of Runx2 which regulates terminal osteogenic differentiation; Msx2, a homeobox-containing transcription factor crucial for cranial bone development, and SRY-box transcription factor 9 (Sox9), a master transcription factor of cartilage differentiation [23,24,25,26]. Osteogenic stimulation of VSMCs and VICs activates specific subsets of these transcription factors that drive the expression of osteogenic markers, including osteocalcin (OCN), bone sialoprotein, and alkaline phosphatase (ALP) [27,28]. Collectively, the coordinated activity of these osteochondrogenic transcription factors facilitates the phenotypic shift in VSMCs from a contractile to an osteoblast-like state, thereby contributing to vascular calcification.

#### 2.2.2. Calcification Inducers

High serum phosphate is a potent inducer of vascular calcification, with high pathological relevance in CKD-associated calcification. Jono et al. demonstrated that elevated phosphate levels promote phosphate uptake in VSMCs via type III sodium-dependent phosphate transporters, particularly PiT-1, triggering osteogenic differentiation of VSMCs and calcific nodule formation [29]. Yang et al. showed that increased extracellular calcium acts synergistically with phosphate to promote matrix mineralization [30]. These ions not only serve as building blocks for hydroxyapatite but also activate key signaling pathways, facilitating the transition of VSMCs to the osteo- and chondrogenic phenotypes. The uremic toxin indoxyl sulfate induces oxidative stress and upregulates osteoblast-specific genes, including Runx2, OCN, and ALP in VSMCs [31].

Type II diabetes is linked to a higher incidence of vascular calcification [32]. Chen et al. demonstrated that elevated glucose levels upregulate Runx2 and its downstream target OCN, increase ALP activity, and promote the secretion of bone morphogenetic protein 2 (BMP-2), a potent osteoinductive factor, in VSMCs [33]. Furthermore, advanced glycation end-products (AGEs) act via receptor for AGE and p38 MAPK pathways to further drive osteogenic signaling [34].

Inflammatory cytokines, including TNF-α, IL-1β, IL-6, and oncostatin M, enhance calcification by inducing ALP in VSMCs [35,36,37]. Oxidized low-density lipoprotein (LDL) and other lipid peroxidation products contribute to VSMC calcification by synergizing with phosphate and enhancing osteoblastic gene expression [38,39]. Lipoprotein (a) has also been shown to induce calcification by inducing the release of calcifying extracellular vesicles from VSMCs and VICs [40].

Recent evidence highlighted the role of hypoxia in cardiovascular calcification. Studies showed that hypoxia is a bona fide osteogenic stimulus that induces VSMC osteochondrogenic differentiation and calcification by increasing the production of reactive oxygen species (ROS) [41]. Moreover, hypoxia synergizes with phosphate to trigger calcification of both VSMCs and VICs [42,43]. Induction of the hypoxia-inducible factor (HIF) pathway plays a critical role in hypoxia-driven calcification. Furthermore, HIF activation by prolyl hydroxylase inhibitors enhances phosphate-induced calcification even under normoxic conditions [43,44].

#### 2.2.3. Calcification Inhibitors

Under physiological conditions, cardiovascular calcification is actively suppressed by a variety of endogenous inhibitors that counterbalance pro-calcific stimuli. Fetuin-A, a liver-derived glycoprotein, plays a pivotal role by binding to nascent calcium phosphate crystals, forming soluble calciprotein particles and thereby preventing their deposition in the vascular wall or the valve [45]. Matrix Gla protein (MGP), a vitamin K-dependent protein expressed by VSMCs, inhibits both BMP-2 signaling and hydroxyapatite crystal formation. Mice lacking MGP develop widespread vascular and cartilaginous calcification, underscoring its critical regulatory role [46]. Another key inhibitor, inorganic pyrophosphate, directly interferes with hydroxyapatite nucleation and elongation [47]. Magnesium also acts as a potent calcification inhibitor by substituting for calcium ions in the crystal structure of hydroxyapatite, leading to the formation of less stable, more soluble mineral phases [48].

## 3. Iron Is a Janus-Faced Modulator of Cardiovascular Calcification

Iron dysregulation is a common feature of several chronic diseases that are associated with accelerated cardiovascular calcification. For example, in CKD, functional and absolute iron deficiency are prevalent due to impaired iron absorption, chronic inflammation, and increased hepcidin levels [49]. On the contrary, the microenvironment of a complicated atherosclerotic plaque is characterized by iron overload, due to intraplaque hemorrhage and the release of heme iron [50]. Therefore, several studies have investigated whether iron plays a role in the calcification of the cardiovascular system. In this chapter, we will discuss the controversial results of these studies.

### 3.1. In Vitro, Ex Vivo, and In Vivo Evidence That Excess Iron Promotes Vascular Calcification

A high iron level, when it exceeds the body’s iron-handling capacity, can be harmful to the vasculature. It contributes to oxidative stress, endothelial damage, lipid peroxidation, and ferroptosis, all of which increase the risk of cardiovascular disease.

Several recent in vitro studies investigated the effect of excess iron on osteogenic differentiation and calcification of VSMCs. Among them, three studies support the hypothesis that iron can boost calcific responses in VSMCs through multiple molecular pathways (Table 1A) [51,52,53].

Kawada et al. demonstrated that exposure of human aortic vascular smooth muscle cells (HAoSMCs) to iron (30–100 µg/mL in the form of holo-ferritin) increased high Pi-induced calcium deposition, particularly in the presence of TNF-α, highlighting a synergy between iron and inflammation [52]. The researchers identified interleukin-24 (IL-24) as a downstream mediator upregulated by iron, which played a crucial role in promoting calcification. Notably, neutralizing IL-24 significantly reduced calcification, underscoring its functional importance in this process [52]. Iron alone and in combination with TNF-α, as well as IL-24, upregulated BMP-2, a well-known pro-osteogenic factor [52]. Ye et al. demonstrated that a high Pi + high calcium-containing osteogenic medium (OM) triggers ferroptosis, a form of iron-dependent lipid peroxidation-mediated cell death, in rat aortic VSMCs [53]. Their data linked the repression of the SLC7A11/glutathione/glutathione peroxidase 4 (GPX4) axis with ferroptosis and the facilitation of calcification. Inhibition of ferroptosis resulted in attenuation of OM-induced calcification of rat aortic VSMCs and aortic rings [53]. Complementing these findings, Aierken et al. investigated the transport mechanisms underlying iron-induced calcification. They found that the zinc–iron transporters SLC39A14 and SLC39A8 played a pivotal role in mediating intracellular iron accumulation in VSMCs [51]. Genetic or pharmacologic inhibition of these transporters attenuated ferroptosis and calcification, confirming that iron uptake pathways are critical mediators of VSMC calcific responses [51]. These findings place iron transporter regulation at the nexus of intracellular iron homeostasis and calcification susceptibility.

In addition to these in vitro investigations, recent studies have highlighted a contributory role of iron in promoting vascular calcification in vivo, particularly through mechanisms involving ferroptosis and oxidative stress (Table 1A). Ye et al. employed a vitamin D3-induced calcification model in C57BL/6 mice, wherein high doses of vitamin D3 triggered medial arterial calcification [53]. They showed that ferroptosis inhibition attenuated mineral deposition within the vessel wall in vitamin D3-overloaded mice [53]. Complementing these findings, Song et al. reported that systemic iron overload, induced by repeated intraperitoneal injections of iron dextran in rats, led to impaired renal function and was strongly associated with vascular calcification in the aorta [54]. Their results indicated that iron accumulation in vascular tissues was accompanied by oxidative stress, increased calcium content, and elevated ALP activity. Iron overload triggered an increase in the expression of calcification-related genes, including Runx2, Bmp2, and Msx-2, further linking iron dysregulation to vascular pathology [54]. Building upon this mechanistic insight, Van den Branden et al. investigated the impact of iron in CKD-associated calcification using an adenine-induced rat model combined with a high phosphate diet to mimic the uremic and pro-calcific environment of human CKD [55]. Repeated intravenous administration of iron sucrose significantly aggravated arterial medial calcification, which was associated with increased iron accumulation in vascular tissues and elevated oxidative stress markers [55]. Together, these studies provide compelling evidence that iron not only contributes to vascular calcification through ferroptosis-mediated VSMC injury and oxidative stress but also amplifies calcification in disease models characterized by disturbed mineral metabolism.

### 3.2. In Vitro, Ex Vivo, and In Vivo Evidence That Excess Iron Inhibits Vascular Calcification

Despite the large body of evidence presented in the previous section, the role of iron in vascular calcification remains far from settled. Intriguingly, several studies have reported an opposite, inhibitory effect of iron against calcification, suggesting a dose-dependent, context-specific, or compartmentalized role for iron in vascular biology. In this section, we turn to this contrasting body of literature, highlighting studies in which iron supplementation or iron-dependent pathways have been shown to inhibit or attenuate vascular calcification (Table 1B).

The first study delivered by Zarjou et al. investigated the effect of heme on osteochondrogenic differentiation and calcification. Surprisingly, heme inhibited high Pi-induced HAoSMC calcification by inhibiting osteoblastic differentiation and reducing the expression of pro-calcific markers such as Runx2 and ALP [56]. The inhibitory effect of heme is attributed to the activation of heme oxygenase-1 (HO-1), the release of heme iron, and the upregulation of ferritin H (FtH), a ferroxidase activity-bearing subunit of ferritin [56]. The pivotal role of ferroxidase activity in inhibiting calcification is underscored by findings showing that a ferroxidase-deficient mutant of FtH is unable to prevent calcification, while ceruloplasmin, a plasma protein possessing ferroxidase activity, effectively reduces calcification [56]. Building upon this concept, Becs et al. showed that pharmacological induction of ferritin effectively prevents the osteogenic reprogramming of VSMCs in vitro. Their study confirmed that ferritin maintains the contractile VSMC phenotype and limits the expression of osteogenic transcription factors under pro-calcific conditions [57].

Beyond endogenous ferritin, iron citrate has emerged as a promising pharmacological agent inhibiting vascular calcification. In a series of studies, Ciceri and colleagues demonstrated that iron citrate prevents Pi-induced VSMC calcification via multiple mechanisms, including the inhibition of high Pi-induced apoptosis, the attenuation of high Pi-induced elevation of the expression of osteogenic and chondrogenic extracellular matrix proteins, thereby preventing the phenotypic switch of VSMCs, and activation of autophagy [58,59,60].

Furthermore, the effect of iron was studied in an ex vivo aorta culture model. Wang et al. showed that iron sucrose prevents high Pi-induced calcification of rat aortic rings [61]. Interestingly, a high concentration of iron sucrose attenuates high Pi-induced ROS production and decreases Pi-induced upregulation of osteogenic markers, including Runx2 and the phosphate transporter PiT-1, while restoring the level of the contractile smooth muscle marker α-SMA [61].

Rajendran et al. investigated the relationship between iron and calcification in a rabbit model of atherosclerosis. Using elemental imaging and histological analysis in atherosclerotic lesions, they demonstrated that regions of iron accumulation were negatively correlated with areas of calcification [62]. Seto et al. explored the effects of iron overload in a uremic rat model and reported a marked reduction in aortic calcification in animals receiving high-dose iron [63]. Histological analyses confirmed reduced calcium deposition and downregulation of osteogenic markers Runx2 and PiT-1 [63]. In contrast, a recent study by Nakanishi et al. found that intraperitoneal iron administration significantly increased aortic iron content, without affecting aortic calcium or phosphate deposition in a rat CKD model [66].

Phosphate binders are a cornerstone of CKD-associated mineral and bone disorder management. Unfortunately, the traditional calcium-based phosphate binders may contribute to ectopic calcification, prompting the development of iron-based alternatives.

Phan et al. compared the effect of a traditional phosphate binder, CaCO_3_, and a novel iron-based phosphate binder, PA21 (sucroferric oxyhydroxide), in rats with CKD. Both CaCO_3_ and PA21 effectively reduced serum phosphate levels, but PA21 was superior in preventing the development of vascular calcification [64]. The authors proposed that this dual benefit arises from both phosphate chelation and direct vascular effects of iron, including potential inhibition of osteoblastic transdifferentiation of VSMCs [64]. Further support comes from Neven et al., who showed that PA21 treatment in CKD rats conferred renoprotective and vasculoprotective effects. Treated animals displayed improved renal function, reduced vascular calcium deposition, and decreased expression of Runx2 [65].

### 3.3. The Potential Role of Iron in Promoting Valve Calcification

Accumulating evidence suggests that iron accumulates in the calcified aortic valve, and studies have proposed that excess iron plays a pro-calcific role in the progression of calcific aortic valve disease (Table 2A). Laguna-Fernandez et al. demonstrated that iron accumulation is more prevalent in calcified aortic valves than in non-calcified tissue [67]. Stam et al. performed a histopathological study analyzing surgically explanted valves from patients with severe aortic or mitral valve disease. They found that intra-leaflet hemorrhage is a frequent and under-recognized feature, especially in symptomatic cases. The hemorrhages were often associated with angiogenesis, microvascular leakage, and calcification. The authors propose that intra-leaflet hemorrhage may be a contributor to local tissue damage and degeneration, possibly initiating or exacerbating processes such as calcification and fibrosis through iron-related oxidative stress and inflammation [68]. In a following study, Morvan et al. analyzed human aortic valve leaflets and found a spatial overlap between iron and calcium deposits, particularly but not exclusively in areas of neovascularization and hemorrhage [69]. They also demonstrated that exposure of VICs to senescent red blood cells (RBCs) in vitro promotes VIC differentiation toward an osteoblast-like phenotype, characterized by increased expression of osteogenic markers, including OPG, BMP-2, MSX-2, as well as elevation of pro-inflammatory cytokines such as IL-1β and IL-6 [69]. However, this study did not delineate which component of the senescent RBCs is responsible for the pro-calcification effect. Xu et al. investigated the effect of iron on the osteogenic differentiation of VICs. This study revealed that iron promotes osteogenic differentiation of VICs only if the cells are deficient in the cystine/glutamate antiporter SLC7A11, a key regulator of ferroptosis resistance [70]. In this context, iron overload increased lipid peroxidation and Runx2 expression in SLC7A11 VICs [70]. Qin et al. confirmed the pro-calcific effect of senescent RBCs and found that iron is capable of inducing VIC calcification through inducing ferroptosis [71].

### 3.4. The Potential Role of Iron in Inhibiting Valve Calcification

Although a spatial link between calcium deposits and intra-leaflet iron accumulation is well-established, the potential causative role of excess iron in valve calcification remained unconfirmed. Interestingly, multiple studies indicate that iron may inhibit, rather than promote, the osteogenic differentiation and calcification of VICs, primarily through antioxidant and cytoprotective mechanisms (Table 2B). Sikura et al. investigated the role of FtH, the ferroxidase-active subunit of ferritin, in the regulation of calcification in VICs. They found that recombinant FtH significantly inhibited phosphate-induced VIC calcification [72]. This inhibitory effect was associated with reduced expression of osteogenic markers such as Runx2 and BMP-2, and decreased oxidative stress [72]. The study also showed that silencing endogenous FtH enhanced mineralization, suggesting a protective, cell-intrinsic role for FtH [72]. The findings indicated that FtH may prevent valve calcification by modulating iron homeostasis and redox balance, as well as by preserving the non-osteogenic phenotype of VICs [72]. Furthermore, Balogh et al. reported that the iron-containing heme potently inhibits high Pi-induced osteogenic phenotype switch and calcification of VICs through the activation of the Nrf2/HO-1 antioxidant pathway [73].

Bioprosthetic heart valves (BHVs) are prepared from porcine heart valves or bovine pericardium and used in heart valve replacement surgeries. The advantage of BVH use over mechanical valves is that patients with BVHs do not require anticoagulation; however, structural valve deterioration or calcification may occur over time, necessitating a second valve replacement surgery [75]. Carpentier et al. explored the use of iron salts as a pretreatment to prevent calcification in bioprosthetic valve tissues [70]. Iron-treated bovine pericardial and porcine aortic tissues exhibited reduced calcium accumulation after subdermal implantation in rats [74]. The observed anti-calcific effect was attributed to iron’s ability to interfere with calcium phosphate crystallization [74]. However, the study also highlighted a critical limitation: iron leaching from the tissue over time, which could reduce the long-term efficacy of iron pretreatment.

### 3.5. The Role of Iron-Loaded Macrophages in Intimal Calcification

Macrophages are key cellular components of atherosclerotic lesions, playing central roles in all stages of atherogenesis, including vascular calcification [76]. Within the atherosclerotic microenvironment, macrophages are exposed to oxidized lipids, cytokines, and hypoxic stress that drive their differentiation into distinct functional phenotypes [77,78]. Classically activated (M1) macrophages adopt a pro-inflammatory profile, producing cytokines like TNF-α, IL-1β, and ROS that promote tissue injury and plaque instability [79]. In contrast, alternatively activated (M2) macrophages exhibit anti-inflammatory and reparative functions, secreting IL-10, TGF-β, and factors that support tissue remodeling and resolution of inflammation [80].

However, macrophage phenotypes in atherosclerosis are not limited to this simple M1/M2 dichotomy; additional subtypes, including Mhem and M(Hb) macrophages, arise in response to heme or hemoglobin exposure within areas of intraplaque hemorrhage [81,82,83,84,85]. These macrophages are primarily involved in heme and iron detoxification, and they display a transcriptional profile distinct from classical M1 and M2 macrophages. Upon engulfing hemoglobin-haptoglobin or heme-hemopexin complexes via CD163 and CD91 receptors, respectively, these macrophages upregulate heme oxygenase-1 (HO-1), which catalyzes the degradation of heme into biliverdin, carbon monoxide, and ferrous iron [86]. To prevent iron-induced oxidative stress, they sequester excess iron in ferritin and export it through ferroportin, maintaining low intracellular labile iron levels [87]. This efficient iron export reduces ROS generation and lipid peroxidation, thereby limiting oxidative damage within the plaque microenvironment. Besides these properties, these macrophages exhibit a distinct atheroprotective phenotype, characterized by increased expression of liver X receptor, which promotes cholesterol efflux and inhibits foam cell formation [84].

Interestingly, a recent study by Sakamoto et al. showed that CD163+ M(Hb) macrophages prevent vascular calcification [88]. They demonstrated a strong inverse correlation between CD163+ macrophages and vascular calcification in human carotid artery advanced atheromas. A deficiency of CD163 in the apolipoprotein E-deficient mouse background enhanced plaque calcification. They showed that supernatant from CD163+ macrophages attenuated VSMC calcification in vitro through a mechanism involving stimulation of NF-κB signaling and enhanced production of the anticalcific extracellular matrix glycosaminoglycan, hyaluronan [88].

## 4. Key Cellular Mechanisms Underlying the Modulatory Effect of Iron on Cardiovascular Calcification

As outlined in the previous chapter, the effect of excess iron on calcification is controversial. However, multiple studies have uncovered distinct cellular pathways through which iron may promote or inhibit calcification. In this chapter, we explore these pathways in detail.

### 4.1. ROS Production and Activation of the Nuclear Factor Erythroid 2-Related Factor 2 (Nrf2) Antioxidant Pathway

Iron is a well-established contributor to intracellular ROS generation, primarily through its involvement in the Fenton and Haber-Weiss reactions. In the Fenton reaction, ferrous iron (Fe^2+^) reacts with hydrogen peroxide (H_2_O_2_) to generate hydroxyl radicals (•OH), one of the most reactive and damaging ROS, along with ferric iron (Fe^3+^). The Haber-Weiss reaction involves the interaction of superoxide anion (O_2_•^−^) with hydrogen peroxide, facilitated by the catalytic cycling of iron between its ferrous and ferric states [89].

Disrupted redox balance and excessive ROS generation are closely linked to various vascular pathologies, including vascular calcification [90,91]. Supporting this association, elevated ROS levels have been observed in vivo within the calcifying aortas of CKD rats, as well as around calcified lesions in the aortic valves of rabbits fed a cholesterol- and vitamin D-rich diet [92,93]. A growing body of evidence suggests that increased ROS production promotes the osteochondrogenic phenotype switch of VSMCs. For instance, H_2_O_2_ and the superoxide-generating xanthine/xanthine oxidase system have both been shown to enhance Pi-induced osteochondrogenic differentiation of VSMCs in vitro [94,95]. Additionally, recent findings have demonstrated that hypoxia, a local condition observed in both intimal and medial calcification, promotes the phenotypic transition of VSMCs into osteochondrogenic cells through a distinctly ROS-dependent mechanism [41]. Based on this evidence, it is reasonable to suggest that elevated iron levels may facilitate VSMC calcification by enhancing ROS generation; however, this possibility has never been experimentally tested.

On the other hand, the redox-active labile form of iron can activate the Nrf2 pathway, primarily through oxidative stress signaling. Nrf2 is a transcription factor and a central regulator of inducible cellular defense mechanisms that enhance the cell’s ability to neutralize and eliminate harmful substances [96]. It plays a critical role in maintaining cellular homeostasis by controlling the expression of more than 250 genes involved in antioxidant defense and phase II detoxification. Under normal conditions, Nrf2 is retained in the cytoplasm by Kelch-like ECH-associated protein 1 (Keap1), which targets it for proteasomal degradation [97]. However, during oxidative stress, modifications to Keap1 disrupt this interaction, allowing Nrf2 to escape degradation and translocate into the nucleus [97]. There, Nrf2 binds to antioxidant response elements in the DNA, triggering the transcription of a range of cytoprotective genes [96]. The major Nrf2-regulated cytoprotective genes include catalase, superoxide dismutase, NAD(P)H:quinone oxidoreductase-1 (NQO1), glutathione S-transferase, glutamate/cysteine ligase, thioredoxin, and glutathione peroxidase. Besides these, Nrf2 targets key proteins involved in iron homeostasis, including both subunits of ferritin (FtH and FtL), the major intracellular iron storage protein, heme oxygenase-1, an inducible enzyme for heme degradation, and ferroportin, the only known iron exporter to date [98]. This mechanism thus represents a double-edged cellular strategy aimed at counteracting iron-induced ROS overproduction and preserving redox balance by simultaneously reducing intracellular labile iron levels and promoting the elimination of ROS.

A growing body of research highlights the protective role of the Nrf2 signaling pathway in vascular calcification, primarily through its ability to counteract oxidative stress and attenuate osteogenic transdifferentiation of VSMCs. Ha et al. showed that pharmacological activation of Nrf2 using dimethyl fumarate (DMF) significantly reduced calcium deposition in rat aortas and cultured VSMCs [99]. The authors demonstrated that DMF enhanced the expression of antioxidant enzymes such as NQO1 and HO-1, which mitigated oxidative stress and downregulated osteogenic markers [99]. Similarly, Zhang et al. showed that resveratrol, a polyphenolic compound, ameliorated vascular calcification in a rat model by upregulating Sirtuin 1 and activating Nrf2, resulting in decreased levels of Runx2 and ALP, key drivers of osteogenic conversion in VSMCs [100]. In another study, Aghagolzadeh et al. demonstrated that hydrogen sulfide attenuated phosphate-induced VSMC calcification through activation of the Keap1/Nrf2/NQO1 axis [101]. Their findings suggested that hydrogen sulfide modulates oxidative stress and protects against calcification by promoting the nuclear translocation of Nrf2 and upregulating cytoprotective genes [101]. Ji et al. reported similar anti-calcific effects with rosmarinic acid, a natural antioxidant compound, which inhibited calcium deposition and osteogenic gene expression in vitro via Nrf2 pathway activation, and these effects were abrogated when Nrf2 was silenced [102]. Building on these observations, Wei et al. provided further mechanistic insight by demonstrating that activation of the Keap1/Nrf2/p62 pathway reduced phosphate-induced VSMC calcification by lowering intracellular ROS levels. This study highlighted a feedback mechanism whereby p62 enhances Nrf2 activation, further strengthening the cell’s antioxidant defense [103]. Cui et al. investigated the effects of mitoquinone, a mitochondria-targeted antioxidant, and found that it alleviated vascular calcification in vitro and in vivo by suppressing oxidative stress and apoptosis in VSMCs through the Keap1/Nrf2 pathway [104]. These collective findings emphasize that Nrf2 serves as a key regulatory mechanism linking redox balance to calcification control, and its targeted activation represents a promising therapeutic strategy against vascular mineralization.

Mounting evidence suggests that iron may exert anti-calcific effects through the activation of the Nrf2 signaling pathway and the upregulation of ferritin, particularly the FtH subunit, which plays a key role in iron sequestration and oxidative stress defense. Zarjou et al. first demonstrated that both exogenous and endogenously expressed ferritin significantly inhibit phosphate-induced calcification and osteoblastic transdifferentiation of VSMCs, primarily by reducing ALP activity and downregulating osteogenic markers such as Runx2 and OCN [56]. Building on this, Becs et al. showed that pharmacological induction of ferritin using iron compounds or 3H-1,2-dithiole-3-thione prevented the osteoblastic transformation of VSMCs and reduced calcium deposition, implicating the Nrf2-ferritin axis as a key modulator of calcification [57]. Sikura et al. extended these findings to valvular cells, demonstrating that FtH attenuates osteogenic differentiation and mineralization in aortic VICs, further supporting the role of ferritin as an endogenous inhibitor of pathological mineralization [72]. Importantly, Balogh et al. provided mechanistic insight by showing that heme, a pro-oxidant iron-containing molecule, activates the Nrf2/HO-1 pathway, which in turn upregulates ferritin expression and reduces calcification in VICs [73]. Notably, inhibition of HO-1 or Nrf2 abolished the anti-calcific effect of heme, confirming that this protective response was mediated via the Nrf2/HO-1 pathway [73]. Moreover, knockdown of FtH reversed the inhibitory effect of heme on VIC calcification, highlighting ferritin as a key downstream effector in this pathway [73]. Collectively, these studies support the hypothesis that iron, through moderate redox activation of Nrf2 and subsequent induction of ferritin, acts as a negative regulator of vascular and valvular calcification by mitigating oxidative stress and inhibiting osteogenic signaling.

### 4.2. Ferroptosis

Ferroptosis is a regulated, non-apoptotic form of cell death characterized by the iron-dependent accumulation of lipid peroxides, resulting in membrane damage and cell death [87,105]. Ferroptosis is driven by oxidative stress resulting from the failure of the cellular antioxidant system, particularly the depletion of glutathione (GSH) and the inactivation of glutathione peroxidase 4 (GPX4), an essential enzyme that reduces lipid hydroperoxides to non-toxic lipid alcohols [106]. Iron plays a central role in this process by catalyzing the Fenton reaction, which generates ROS that promote lipid peroxidation. Key regulators of ferroptosis include the cysteine/glutamate antiporter system, particularly SLC7A11, the key subunit that is responsible for importing cysteine for GSH synthesis, GPX4, and iron-handling proteins such as ferritin and transferrin receptor 1. Ferroptosis has been implicated in a variety of pathological conditions, including neurodegeneration, cancer, ischemia–reperfusion injury, and, more recently, vascular and renal diseases.

Several recent studies highlight that ferroptosis contributes to VSMC osteogenic transdifferentiation and calcification through diverse regulatory axes. Central to this process is the repression of the SLC7A11/GPX4 axis, which impairs cellular antioxidant defenses, triggering ferroptotic death and calcification of VSMCs [53]. Epigenetic regulators such as histone methyltransferase G9a and histone deacetylase 9 exacerbate this process by transcriptionally suppressing SLC7A11 [107,108], while p53 activation further enforces ferroptotic signaling via SLC7A11 downregulation [109]. The clinical relevance of SLC7A11 was underscored in a prospective observational cohort study in which the authors measured serum SLC7A11, IL-6, IL-1β, and C-reactive protein levels and assessed abdominal aortic calcification scores. They found that patients with moderate to severe vascular calcification had significantly lower serum SLC7A11 levels. Serum SLC7A11 levels showed a negative correlation with phosphate, calcium, and IL-1β. In multivariate analysis, dialysis duration, SLC7A11, and phosphate levels emerged as independent risk factors associated with vascular calcification. Furthermore, low SLC7A11 levels were associated with a poorer clinical prognosis over a one-year follow-up period [110].

Ferritinophagy, a form of selective autophagy that involves the degradation of ferritin, increases the availability of redox-active iron, thereby sensitizing cells to ferroptosis, has also been implicated in the process of vascular calcification. Wang et al. showed that high Pi increased lipocalin 2 expression in VSMCs that interacted with nuclear receptor coactivator 4, accelerating the degradation of FtH and inducing ferroptosis [111].

### 4.3. Iron–Phosphate Complex Formation

Iron may exert protective effects against vascular and bioprosthetic calcification through its capacity to form soluble iron–phosphate complexes, thereby reducing the pool of free inorganic phosphate available for pathological mineralization. Vasudev et al. demonstrated that iron ions, in conjunction with magnesium, significantly inhibited calcification in polyethylene glycol-modified bovine pericardium, with the proposed mechanism involving direct chemical interaction between iron and phosphate, preventing hydroxyapatite nucleation [112]. Similarly, Carpentier et al. reported that iron treatment reduced bioprosthetic valve mineralization, suggesting a physicochemical mechanism independent of cellular modulation [74]. Supporting these findings, Hilton et al. showed that phosphate can bind ferric iron in solution to form soluble Fe(III)-phosphate complexes, which may act as non-transferrin-bound iron species, limiting iron incorporation into transferrin and altering systemic iron distribution [113]. Additionally, Halliwell and Gutteridge highlighted that the catalytic reactivity of iron is profoundly influenced by its chemical speciation, and complexation with phosphate may reduce its redox activity and availability for participation in mineral deposition [114]. Collectively, these studies suggest that iron’s ability to sequester phosphate into stable, non-crystalline complexes may represent a non-cellular mechanism of calcification inhibition, particularly relevant in diseases with hyperphosphatemia such as CKD.

## 5. Iron Dysregulation in CKD and Its Association with Vascular Calcification

Anemia is one of the most common and clinically significant complications of CKD, with prevalence increasing as kidney function declines. In a large U.S. population study, more than half of patients with advanced CKD were found to be anemic, highlighting the substantial burden of this condition in affected individuals [115]. The pathogenesis of anemia in CKD is multifactorial, involving reduced erythropoietin production, iron deficiency, chronic inflammation, and the dysregulation of iron metabolism, which is closely linked to disturbances in phosphate and mineral homeostasis [116]. Iron deficiency anemia, in particular, is highly prevalent in CKD and is driven by reduced dietary iron intake, impaired intestinal iron absorption, blood loss, and increased hepcidin levels that limit iron availability for erythropoiesis [117]. Low serum iron, reduced ferritin, low transferrin saturation (TSAT), and elevated transferrin levels are characteristic markers of iron deficiency. In CKD, iron deficiency is generally defined by a serum ferritin level below 100 ng/mL and TSAT below 20%.

The clinical importance of anemia in CKD is underscored by its association with reduced quality of life, increased cardiovascular risk, and higher mortality, as demonstrated in both epidemiological and primary care studies [118]. Iron deficiency anemia is commonly managed with erythropoiesis-stimulating agents and iron supplementation [4].

Vascular calcification is a characteristic complication of chronic kidney disease and is particularly prevalent among patients undergoing maintenance hemodialysis, where it strongly predicts cardiovascular morbidity and mortality. The effects of iron on VSMC calcification have been described, which initiated studies to reveal the relationships between iron, carotid artery calcification, and mortality among CKD patients. The study by Mizuiri and colleagues provides valuable insight into this relationship. In a cohort of 173 patients receiving maintenance hemodialysis, coronary artery calcification was quantified and related to iron parameters, including serum iron, transferrin saturation, and ferritin. Patients with severe calcification, defined as a coronary artery calcification score of 400 or higher, exhibited significantly lower serum iron and transferrin saturation. A transferrin saturation above 21% was independently associated with a reduced likelihood of severe calcification, suggesting that functional iron deficiency contributes to the development of vascular calcification in dialysis patients. Moreover, over a five-year follow-up, higher transferrin saturation, serum iron, and ferritin levels were all independently associated with improved survival, demonstrating that markers of adequate iron stores predict better long-term outcomes in this patient population [5]. More recently, Mizuiri et al. specifically focused on the relationship between absolute iron deficiency and coronary artery calcification in maintenance hemodialysis patients. Their study demonstrated that patients with low transferrin saturation and low ferritin not only had higher coronary artery calcification scores but also experienced increased cardiovascular mortality over a three-year follow-up period, emphasizing the clinical significance of iron deficiency as a predictor of vascular outcomes [119]. In contrast, Chen et al. found that elevated ferritin heavy chain expression was independently associated with the development and progression of coronary artery calcification [120].

## 6. Concluding Remarks

Vascular calcification remains a formidable challenge in CKD, particularly in patients on maintenance hemodialysis, where it contributes substantially to cardiovascular morbidity and mortality. Evidence accumulated over the past decade indicates that iron status is a significant and previously underappreciated determinant of calcification risk. Observational studies consistently link both functional and absolute iron deficiency with increased coronary artery calcification and decreased survival. Experimental studies underscore the dual nature of iron in vascular calcification (Figure 1). While iron excess has been shown to amplify oxidative stress, lipid peroxidation, and ferroptotic cell death, thereby accelerating medial calcification and osteogenic reprogramming, other findings indicate that excess iron can also exert protective effects. The anti-calcification effect of iron relies on strengthening the cellular antioxidant defense mechanisms and reducing free phosphate availability via the formation of soluble iron–phosphate complexes. Such divergent observations suggest the context-dependent role of iron in vascular biology and warrant the need for careful iron management in clinical practice. 

Maintaining iron homeostasis is crucial not only for treating CKD-associated anemia but also for preventing vascular calcification in dialysis populations. However, the current evidence base is largely associative, and causality remains to be established. Interventional trials that directly test iron-based strategies with vascular calcification and cardiovascular outcomes as endpoints are urgently needed. The optimal range of iron parameters that balances anemia and provides vascular protection needs to be established and introduced to clinical practice. Until then, careful monitoring of iron status, avoidance of both deficiency and excess, and integration of mechanistic insights into patient management represent the most rational approach.

## Figures and Tables

**Figure 1 ijms-26-10210-f001:**
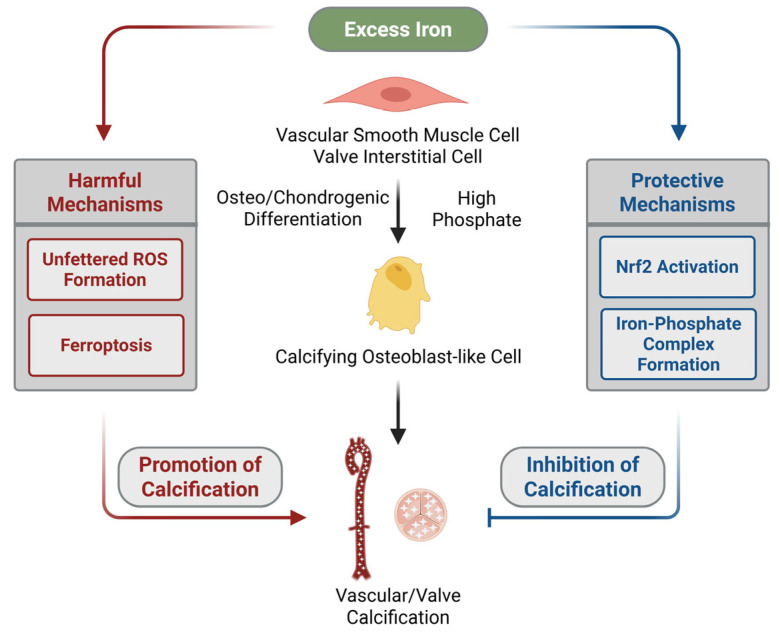
The dual role of excess iron in cardiovascular calcification. Excess iron modulates the osteo/chondrogenic differentiation of vascular smooth muscle cells and valve interstitial cells. On one hand, iron can promote calcification by increasing reactive oxygen species (ROS) production or inducing ferroptosis. On the other hand, it may exert protective effects through activation of the cytoprotective Nrf2 pathway or formation of iron–phosphate complexes, thereby inhibiting vascular and valvular calcification.

**Table 1 ijms-26-10210-t001:** Studies addressing the effect of iron on vascular calcification.

**A. Iron Promotes Vascular Calcification**			
	Experimental Model	Major Finding	Reference
**In vitro/ex vivo studies**	Cell type: HAoSMCs Calcification induction: high Pi+TNF-alpha Treatment: iron overload: holoferritin	Excess iron accelerates Pi-, and Pi+TNF-α-induced calcification of HAoSMCs through IL-24 upregulation.	Kawada et al., 2018 [52]
	Cell type: primary rat VSMCs/rat aortic rings Calcification induction: high Ca+high Pi Treatment: ferroptosis inhibition: Ferrostatin-1	Inhibition of ferroptosis with Ferrostatin-1 ameliorates rat VSMC and aortic ring calcification.	Ye et al., 2022 [53]
	Cell type: primary mouse VSMCs Calcification induction: β-GP+high Ca Treatment: iron overload: Iron(II) ammonium citrate; iron deficiency: Desferrioxamine	FAC promotes, and DFO inhibits VSMC osteogenic differentiation and calcification.	Aierken et al., 2024 [51]
**In vivo studies**	Model: vitamin D3-overloaded mice Treatment: ferroptosis inhibition: Ferrostatin-1	Inhibition of ferroptosis with Ferrostatin-1 ameliorates aortic calcification.	Ye et al., 2022 [53]
	Model: iron overload rats Treatment: iron-dextrane, i.p.	Iron treatment triggers iron accumulation and increases arterial calcification.	Song et al., 2022 [54]
	Model: adenine-induced CKD in rats Treatment: iron-sucrose, i.v.	Iron treatment triggers iron accumulation and lipid peroxidation, and increases CKD-induced arterial calcification.	Van den Branden et al., 2025 [55]
**B. Iron Inhibits Vascular Calcification**			
	Experimental Model	Major Finding	Reference
**In vitro/ex vivo studies**	Cell/tissue type: HAoSMCs Calcification induction: high Pi Treatment: iron overload: heme/iron chloride/ferritin; ferroxidases: H ferritin, ceruloplasmin	Excess iron inhibits Pi-induced calcification of HAoSMCs through the upregulation of ferritin. Ferroxidase activity provides the calcification-inhibitory effect.	Zarjou et al., 2009 [56]
	Cell/tissue type: HAoSMCs Calcification induction: β-GP+vitamin D3 Treatment: iron overload: ferritin; ferritin H induction: 3H-1,2-Dithiole-3-thione	Induction of ferritin prevents osteoblastic transformation and calcification of HAoSMCs.	Becs et al., 2016 [57]
	Cell/tissue type: primary rat VSMCs Calcification induction: high Pi Treatment: iron overload: iron(III) citrate	Excess iron inhibits high Pi-induced VSMC calcification by preventing apoptosis, inducing autophagy, and affecting osteoblastic differentiation.	Ciceri et al., 2016 and 2019 [58,59,60]
	Cell/tissue type: rat aortic ring Calcification induction: high Pi Treatment: Iron overload: iron(III) sucrose	Excess iron inhibits high Pi-induced aortic ring calcification and osteogenic differentiation of VSMCs.	Wang et al., 2021 [61]
**In vivo studies**	Model: high cholesterol-induced atherosclerosis in rabbits	Calcium and iron levels show an inverse correlation in atherosclerotic lesions.	Rajendran et al., 2012 [62]
	Model: adenine-induced CKD in rats Calcification enhancement: Pi-enriched diet Treatment: iron dextran, i.p.	Reduction in aortic calcification and downregulation of osteogenic markers Runx2 and PiT-1 in animals receiving high-dose iron.	Seto et al., 2014 [63]
	Model: adenine-induced CKD in rats Calcification enhancement: Pi-enriched diet Treatment: CaCO_3_ or iron-based phosphate binder (PA21)	Both CaCO_3_ and PA21 effectively reduced serum phosphate levels, but PA21 was superior in preventing vascular calcification.	Phan et al., 2013 [64]
	Model: adenine-induced CKD in rats Calcification enhancement: Pi-enriched diet Treatment: iron-based phosphate binder (PA21)	PA21 treatment improved renal function, reduced vascular calcium deposition, and decreased expression of Runx2.	Neven et al., 2020 [65]

**Table 2 ijms-26-10210-t002:** Studies addressing the effect of iron on valve calcification.

**A. Iron Promotes Valve Calcification**			
	Experimental Model	Major Finding	Reference
**In vitro studies**	Cell type: human VICs Treatment: senescent RBCs	Senescent RBCs promote VIC differentiation toward an osteoblast-like phenotype.	Morvan et al., 2019 [69]
	Cell type: Slc7a11-deficient human VICs Calcification induction: osteogenic differentiation medium Treatment: iron overload: Iron(II) sulphate	Excess iron promotes osteogenic differentiation of Slc7a11-deficient VICs.	Xu et al., 2022 [70]
	Cell type: primary mouse VSMCs Calcification induction: β-GP+high Ca Treatment: iron overload: Iron(III) ammonium citrate; iron deficiency: Desferrioxamine	Excess iron promotes, and low iron inhibits VSMC osteogenic differentiation and calcification.	Qin et al., 2025 [71]
**In vivo observations**	Model: human aortic valve leaflets	Iron accumulation is more prevalent in calcified valves than in non-calcified tissues. Iron-containing valve regions show increased expression of genes involved in calcification.	Laguna-Fernandez et al., 2016 [67]
	Model: human aortic valve leaflets	There is a spatial overlap between iron and calcium deposits in human aortic valve leaflets.	Morvan et al., 2019 [69]
	Model: human aortic valve leaflets	Intra-leaflet hemorrhages are often associated with angiogenesis, microvascular leakage, and calcification.	Stam et al., 2017 [68]
**B. Iron Inhibits Valve Calcification**			
	Experimental Model	Major Finding	Reference
**In vitro/ex vivo studies**	Cell/tissue type: human VICs Calcification induction: high Pi Treatment: iron overload: heme/iron chloride/ferritin; ferroxidases: H ferritin, ceruloplasmin	Iron-mediated FtH upregulation inhibits high Pi-induced VIC calcification through reduced expression of osteogenic markers such as Runx2 and BMP-2, and decreased oxidative stress.	Sikura et al., 2019 [72]
	Cell/tissue type: human VICs Calcification induction: high Pi Treatment: iron overload: heme/iron(II) chloride/ferritin; Nrf2 and HO-1 inhibition	Heme inhibits high Pi-induced osteogenic phenotype switch and calcification of VICs through the activation of the Nrf2/HO-1 antioxidant pathway.	Balogh et al., 2021 [73]
**In vivo studies**	Model: glutaraldehyde-pretreated porcine bioprosthetic heart valve tissue implanted in rats subdermally Calcification induction: Warfarin Treatment: iron overload: pretreatment with iron(III) nitrate	Iron pretreatment attenuates calcium accumulation in the implanted bioprosthetic heart valve tissue.	Carpentier et al., 1995 [74]

## Data Availability

No new data were created or analyzed in this study.

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
