# Peer review of "Iron in Vascular Calcification: Pro-Calcific Agent or Protective Modulator?"

_ijms, 2025, doi:10.3390/ijms262010210_

Round 1
Reviewer 1 Report
Comments and Suggestions for Authors
The manuscript by Balogh et al. reviews the current knowledge on the role of iron in vascular calcification. The authors describe an important number of studies supporting the concept of the role of iron in vascular pathology and the therapeutic importance of their pharmacological regulation. The review is well concepted and written. There are a few minor concerns, which the authors should address.
- Which iron levels are considered to be pathological in the context of calcification?
- Are there new biomarkers for iron-related vascular disease, for example for ferroptosis?
- What are the new therapeutic strategies to regulate high/low iron levels?
- A graphical abstract or a figure, describing the main pathological and signalling ways is required.
Author Response
Thank you to the reviewer for the valuable comments. Here is a detailed response to each of the points.
Comment 1: Which iron levels are considered to be pathological in the context of calcification?
Reply: We have completed the introduction with these informations and inserted a new reference as follows: „More focused clinical studies have now reported associations between iron markers and measures of coronary calcification: lower serum iron (Fe ≤ 72 µg/dL) and transferrin saturation (TSAT ≤ 21%) have been associated with higher coronary artery calcium scores (CACS ≥ 400) and with worse long-term survival in hemodialysis cohorts; at the same time, absolute iron deficiency (TSAT < 20%, serum Fe < 60) has also been linked with increased cardiovascular mortality in maintenance dialysis patients [4].”
Comment 2: Are there new biomarkers for iron-related vascular disease, for example for ferroptosis?
Reply: Thank you for the question. Ferroptosis is a new type of iron-dependent programmed cell death. The biological characteristics of ferroptosis include abnormal lipid peroxidation and ROS production; however, these markers are not specific for ferroptosis. SLC39A14 and SLC39A8 are metal transporters which are implicated in iron overload-induced ferroptosis in VSMCs, while GPX4 is a critical regulator of ferroptosis via preventing lipid peroxidation. These biomarkers offer valuable insights into the role of ferroptosis in vascular diseases and may serve as potential targets for therapeutic interventions. However, further research is needed to validate their clinical applicability as biomarkers.
Comment 3: What are the new therapeutic strategies to regulate high/low iron levels?
Reply: Thank you for the question. There are various oral and intravenous iron salts in clinical use for treating CKD-associated anemia. A recent review by Orlando M Gutiérrez summarized this topic in detail; therefore, we added this reference to the paper (ref #4). High iron levels can be treated by phlebotomy or chelation therapy.
Comment 4: A graphical abstract or a figure, describing the main pathological and signalling ways is required.
Reply: Thank you for the suggestion. We have submitted a graphical abstract, but it was not inserted into the manuscript file. Now, in the revised version of the manuscript, we have added it as Figure 1.
Reviewer 2 Report
Comments and Suggestions for Authors
The manuscript (ijms-3894777) is a review article on iron and atherosclerotic vascular calcification. Although this is an interesting topic, there are a number of reviews already published. For instance:
Neven E, De Schutter TM, Behets GJ, Gupta A, D'Haese PC. Iron and vascular calcification. Is there a link? Nephrol Dial Transplant. 2011 Apr;26(4):1137-45.
Ciceri P, Cozzolino M. The emerging role of iron in heart failure and vascular calcification in CKD. Clin Kidney J. 2020 Sep 10;14(3):739-745.
Pan W, Jie W, Huang H. Vascular calcification: Molecular mechanisms and therapeutic interventions. MedComm (2020). 2023 Jan 3;4(1):e200.
My first comment is why these well-cited reviews have not been referenced and discussed, and what new insights readers can gain from this review paper.
Secondly, macrophages play important roles in disordered iron metabolism in atherosclerotic vessel walls, particularly regarding the impact of iron on calcification in atherosclerotic vascular diseases. Findings on macrophages and iron in atherosclerosis should not be neglected in a further review paper.
Thirdly, excess iron in atherosclerotic vessel walls is stored largely in macrophages in the development process of vascular calcification. To conclude that ferritin acts as a negative regulator of vascular calcification, previous studies on ferritin and macrophages in atherosclerosis must be considered.
Lastly, the authors aim to discuss the role of ferroptosis in the process of vascular calcification; however, I find no information on how the concept of ferroptosis has been developed based on previous studies of iron and macrophages in atherosclerotic vessel walls.
Author Response
Thank you to the reviewer for the valuable comments. Here is a detailed response to each of the points.
Comment 1: My first comment is why these well-cited reviews have not been referenced and discussed, and what new insights readers can gain from this review paper.
Reply: Thank you for this comment. We apologize for not citing the previously published reviews in this field, as we initially chose to focus on the original experimental studies. However, in response to the reviewer’s suggestion, we have now cited all relevant reviews in the revised manuscript. Our review aimed to compile studies investigating the role of iron in vascular and valve calcification. We found that the literature presents many contradictory findings—some studies report that iron promotes calcification, while others suggest it inhibits it. We believe that the narrative of our review is novel, as no previous reviews have examined both perspectives side by side. We hope that readers will gain new insights from our work, particularly regarding the complexity and nuanced nature of this topic.
Comment 2: Secondly, macrophages play important roles in disordered iron metabolism in atherosclerotic vessel walls, particularly regarding the impact of iron on calcification in atherosclerotic vascular diseases. Findings on macrophages and iron in atherosclerosis should not be neglected in a further review paper.
Comment 3: Thirdly, excess iron in atherosclerotic vessel walls is stored largely in macrophages in the development process of vascular calcification. To conclude that ferritin acts as a negative regulator of vascular calcification, previous studies on ferritin and macrophages in atherosclerosis must be considered.
Reply to comments 2-3: Thank you for this insightful suggestion. In response, we have added a new section (3.5) entitled “The Role of Iron-Loaded Macrophages in Intimal Calcification.” We hope that this addition provides a comprehensive overview and appropriately cites the relevant literature in this important and rapidly evolving field.
Comment 4: Lastly, the authors aim to discuss the role of ferroptosis in the process of vascular calcification; however, I find no information on how the concept of ferroptosis has been developed based on previous studies of iron and macrophages in atherosclerotic vessel walls.
Reply: We thank the reviewer for this insightful comment. In this paper, we focused on the effects of iron on vascular smooth muscle cells and valve interstitial cells in the context of calcification. While we fully agree that macrophage ferroptosis in the atherosclerotic vessel wall is an important and emerging topic, we feel that it lies somewhat outside the specific scope of the present review. Therefore, in this manuscript, we have limited our discussion to studies in which iron-mediated modulation of calcification is directly linked to ferroptosis in vascular smooth muscle cells.